# Green Synthesis of Platinum Nanoparticles for Biomedical Applications

**DOI:** 10.3390/jfb13040260

**Published:** 2022-11-21

**Authors:** Ekaterina O. Mikhailova

**Affiliations:** Institute of Innovation Management, Kazan National Research Technological University, K. Marx Street 68, 420015 Kazan, Russia; katyushka.glukhova@gmail.com

**Keywords:** platinum nanoparticles, green synthesis, capping agents, antibacterial activity, anticancer activity

## Abstract

The diverse biological properties of platinum nanoparticles (PtNPs) make them ideal for use in the development of new tools in therapy, diagnostics, and other biomedical purposes. “Green” PtNPs synthesis is of great interest as it is eco-friendly, less energy-consuming and minimizes the amount of toxic by-products. This review is devoted to the biosynthesis properties of platinum nanoparticles based on living organisms (bacteria, fungi, algae, and plants) use. The participation of various biological compounds in PtNPs synthesis is highlighted. The biological activities of “green” platinum nanoparticles (antimicrobial, anticancer, antioxidant, etc.), the proposed mechanisms of influence on target cells and the potential for their further biomedical application are discussed.

## 1. Introduction

Since the end of the 20th century, the popularity of metal nanoparticles has grown from year to year. This is not surprising because they are widely used in various spheres of human activity owing to their many valuable properties. Green technologies—eco-friendly technologies—are of particular importance as they are simple, cheap, practically waste-free and possess the ability to control the resulting nanoparticles characteristics (size, shape, stability), as well as being a popular topic to study in recent years. This is confirmed by the impressive dynamics of the number of publications on this topic that have emerged in the last twenty years. [1]. Different nanoparticles, nanocomposites and nanostructures possessing biocompatibility are of interest for a wide variety of human activity fields: in the food industry, food packaging and the development of functional food products to increase food safety, the detection of food pathogens and extend the shelf life of food products [2], as well as catalysts for biofuels (for example, Cs2O−MgO/MPC nanocomposite was used as the main nanocatalyst for the production of biodiesel from olive oil) [3], application in makeup and skin care [4], and, of course, use in diagnosis, medical treatment, theranostics, and tissue engineering (these are not only metal nanoparticles, but also, for instance, nanostructured CaPs with surface-rich–OH groups and Ca^2+^ cations, which can effectively adsorb therapeutic agents [5]). Biosynthesis strategies involve using living objects—microorganisms, fungi or plants—as bio-factories for metal nanoparticles production [6,7,8,9]. The obtained medicinal products reveal wide application prospects for the biomedical field for combating pathogens of various diseases, the prevention and treatment of oncological diseases, drugs delivery, diagnostic systems, etc.

Although metal nanoparticles such as silver and gold are more well known, the synthesis and analysis of other metal nanoparticles are also gaining momentum [10]. One of these metals is platinum. The Incas were the pioneers of its mining and application, but in the Old World, platinum was unknown until the 16th century. It was first introduced to the conquistadors from South America, and received its name from the Spanish word *platina*, literally meaning “little silver” according to its external similarity to real silver [11]. Possessing high refractory, platinum was not found worthy of use for a long time, being valued much lower than silver and gold. Between 1889 and 1960, 90% of platinum alloy was used as the international standard for one meter determining. Currently, South Africa and Russia are the leaders in platinum extraction. Platinum is an inert metal, and apparently does not play an important role in the vital activity of living organisms. In addition, this metal is non-toxic in metallic form.

Platinum finds its application in electroplating. It is used as a catalyst in various industries including coating microwave technology elements as well as jewelry. It is used for medical purposes in dentistry, and platinum compounds, cytostatics such as cisplatin, are applied for oncological disease therapy [12,13,14,15]. However, drugs such as cisplatin and carboplatin have nephrotoxic, neurotoxic and cytotoxic effects [16]. These effects can be neutralized with the help of green synthesis of platinum nanoparticles and become the key to solving many medical problems. Bio-factories (microorganisms, fungi, plants) are full of various cellular compounds such as proteins, enzymes, acids, etc., important for the characteristic features formation of platinum nanoparticles synthesized in different organisms. The bio-compound importance consists of not only their participation in the nanoparticle’s synthesis, but also the final assembly processes. Having their own remarkable properties, they will be able to multiply the positive effect of platinum nanoparticles use. Moreover, the agenda of “green” technologies, the absence of toxic side effects, and the targeted effect on the human body are still relevant. The present available information about PtNPs antibacterial properties [17], antitumor effects [18], and other potentially beneficial features make them an interesting topic for comprehensive research. This review is devoted to the biological synthesis of platinum nanoparticles, the mechanism of this process and its effects on the cells of living organisms, as well as potential applications for biomedical purposes.

## 2. The Proposed Mechanism of PtNPs Synthesis

Among the metallic nanoparticles, platinum is a precious metal, extensively used in various chemical and biological sectors due to its properties such as high surface area, excellent resistance to corrosion and chemical attacks. Metal nanoparticle synthesis involves a variety of the same production approaches as platinum. These receiving and stabilization approaches can be divided into two categories according to the “top-down” and “bottom-up” principles [19,20]. The latter is based on physical methods: nanoparticle production starts with the “fragmentation” of massive material into nanoscale particles. The physical method type includes laser ablation, arc discharge, pyrolysis with flame spraying, ball grinding, melt mixing, etc. In the bottom-up approach, nanoparticles are synthesized by self-assembling atoms into nuclei, later developing into nanoscale particles [10]. As a result of the atom reduction or oxidation, they are transformed into atoms with zero valence and then combined into nanoparticles. Finally, a polymolecular, colloidal system with varying dispersion degrees is formed. Chemical and biological methods for metal nanoparticle production are based on it. Chemical approaches include electrochemistry, chemical reduction and photochemical reduction methods [21]. Despite the apparent advantages, chemical methods also have disadvantages, and the main one is the use of toxic reagents. “Green” biological methods represent a special “evolution branch” of nanoparticle synthesis. Their most important advantages are simplicity, environmental friendliness, biocompatibility, manageability, and cost-effectiveness. The biosynthesis of platinum nanoparticles using unicellular and multicellular living organisms (bacteria, fungi, algae, plants) is also an excellent alternative to chemical and physical methods because of biocompounds (alcohols, ketones, aldehydes, flavones, amides, terpenoids, carboxylic acids), which participate in the final capping of nanoparticles, having biomedical significance and being capable of additional effects (Figure 1).

Although the synthesis mechanism of platinum nanoparticles in living organisms is not fully understood, the main stages are typical for other metal nanoparticles [22]. Green synthesis is possible in two ways: directly in the body of bacteria, fungi, plants, or by extracted bioreagents. In general, the relatively simple process involves mixing a plant extract or bacterial culture with a metal ion solution at a certain temperature and pH, which affects the final shape, size, and morphology of the nanoparticles [20]. The concentration increase in plant extract plays an important role, enhancing nanoparticles synthesis and influencing their morphology [23]. The production of nanoparticles can be indicated by color change [24]. Platinum salts such as H_2_PtCl_6_, K_2_PtCl_6_, K_2_PtCl_4_, PtCl_2_, Pt(AcAc)_2_, Pt(NH_3_)_4_-(OH)_2_, Pt(NH_3_)_4_(NO_3_)_2_, and Pt(NH_3_)_4_Cl_2_ are applied for biosynthesis. The biochemical reduction of platinum salts to Pt^0^ is based on redox reactions realized by natural reducing biopolymers. In addition, these compounds can act not only as reducing agents, but also as colloidal biostabilizers, or capping agents. The reduction process is implemented via various biopolymers: proteins and polysaccharides, as well as alcohols, aldehydes, ketones, acids, biologically active substances and other metabolic products [25]. The PtNPs synthesis was observed in bacteria, fungi, algae, and plants, but more exotic bioreactors, for example, egg yolk, were investigated [26,27]. The hydrogenase enzyme participation was found to promote the chemical reaction of Pt(IV) to PtNPs through an intermediate Pt(II) cation in a two–stage reduction mechanism in sulfate-reducing bacteria [28]. The proposed mechanism is shown in Figure 2. The biosynthesis of platinum nanoparticles can be divided into four stages: reduction in Me^+^ metal ions and the formation of neutral Me^0^ atoms: Pt^4+^→2e Pt^2+^→2e Pt^0^ (with the enzymes, polyphenols, terpenoids and reducing sugars participation) [29]; nucleation of metal atoms with the nanoparticle formation; aggregation: small particle’s combination into larger nanostructures formed at the nucleation stage; and termination (stabilization): thermodynamic equilibrium in a colloidal system [30]. The different biomolecule participation (peptides, enzymes, carboxylic acids, aldehydes, ketones, etc.) in the stabilization of nanoparticles as capping agents, prevention of their agglomeration and packaging in aqueous solution was already proved [31]. The variety of PtNPs shapes (spherical, cubic, flower-shaped, hexagonal and others) and sizes was discovered [32]. Diverse methods are actively used to characterize platinum nanoparticles. FTIR analysis (infrared spectroscopy with Fourier transform) makes it possible to analyze biomolecules involved in the reduction in platinum ions and nanoparticle stabilization [33]. The shape and size of synthesized “green” PtNPs are determined using scanning electron microscopy (SEM) and transmission electron microscopy (TEM), UV/Vis spectrophotometry and dynamic light scattering (DLS) are required for evaluating the physical properties of nanoparticles, scanning electron microscopy (SEM) to assess the nanoparticle morphology. The crystalline size of Pt nanoparticles was calculated based on X-ray diffraction measurements [34,35,36].

*Biosynthesis by bacteria*. Microbial synthesis seems to be one of the easy and most reliable approaches to PtNPs production. It is non-toxic, low-cost, conducted in an aqueous solution and able to produce nanoparticles differing in shape, size and physicochemical properties. Their high growth rate and uncomplicated cultivation make these microorganisms a great choice to obtain platinum nanoparticles. Nevertheless, the number of studies devoted to PtNPs bacterial synthesis is significantly inferior to plant-mediated synthesis (discussed later), most likely due to the various vegetal bio compounds having potential medical significance and participating in the synthesis of nanoparticles. Microorganisms synthesize nanoparticles for protection against the toxicity of heavy metals suppress the function of various enzymes and lead to reactive oxygen species (ROS) formation. Various methods are used for this purpose such as removal from cells using transporters and packing into vacuoles [37]. Some bacteria have the ability to absorb metal ions on their surface and eventually reduce them to the corresponding nanoparticles by means of reductase, cytochrome, and metallothioneins [38]. The neutralized metals are precipitated as minerals or stored in cells as metal ions [39].

Both extracellular and intracellular bacterial synthesis are possible. The intracellular mechanism is performed by the microorganism unique transport systems, where the cell wall plays an important role according to its negative charge: positively charged metal ions are attached to the negatively charged cell walls through electrostatic interactions [35]. After transportation to the cells, ions are reduced by enzymes in the periplasmic space (Figure 2), and at the same time NPS can be removed from the cell and subsequently attached to the cell surface, preventing the re-entry of metal into the periplasm and acting as a catalyst for further metal reduction [40]. Thus, intracellular synthesis was shown through cytochromes and/or hydrogenases enzymes in *Desulfovibrio alaskensis*: the formed nanoparticles are exported to the cell outer surface, thus forming extracellular PtNPs [41]. It was also discovered that sulfate-reducing bacteria are able to reduce platinum (IV) to platinum (0) by the participation of two hydrogenases: cytoplasmic hydrogenase reducing platinum (IV) to platinum (II), and periplasmic hydrogenase reducing to platinum (0) [28,42]. Proteins capable of reducing and stabilizing platinum nanoparticles, forming a colloidally stable solution, were found in *Acinetobacter calcoaceticus* [43]. Enzymatic platinum reduction in the periplasmic space with lactate as the electron donor was shown for *Shewanella algae* [44]. Extracellular synthesis is possible owing to bacterial cell secreted compounds reducing platinum ions to a metal with a zero-oxidation degree, in addition to biomolecules stabilizing the resulting nanoparticles participating in the process [45,46]. The main role usually belongs to enzymes that reduce metal ions [47]. For instance, extracellular synthesis was indicated in *Pseudomonas aeruginosa* [48]. The supposed mechanism of PtNPs extracellular production in *Streptomyces sp*. is associated with the participation of enzyme chloride reductase in the nitrogen cycle, which responsible for the chloride reduction to chlorine [49].

Various supernatant-rich bacterial cell lysate reductase able to synthesize platinum nanoparticles were used in Gram-negative bacteria experiments such as *Pseudomonas kunmingensis*, *Psychrobacter faecalis*, *Vibrio fischeri*, and Gram-positive bacteria *Jeotgalicoccus coquinae*, *Sporosarcina psychrophile* and *Kocuria rosea* [50].

Biomolecules stabilizing bacterial PtNPs play a significant role in the final capping and their properties. Very few studies on this topic were found in the literature, but most of them indicate amino acids, proteins [49,50] and primary and secondary amine involvement as capping agents [48].

*Biosynthesis by fungi.* The platinum nanoparticle synthesis was revealed in *Fusarium oxysporum* either on the cell wall/membrane or extracellularly [51]. The proposed mechanism is based on platinum ions reduction via hydrogenase [52]. Octahedral H_2_PtCl_6_ is too large to fit into the active region of the enzyme and, under conditions optimum for nanoparticle formation (pH 9, 65 °C), undergoes a two-electron reduction to PtCl2 on the molecular surface of the enzyme. This smaller molecule is transported through hydrophobic channels within the enzyme to the active region, where it undergoes a second two-electron reduction to Pt(0) [52]. Capping agents are amides as well as proteins that can bind to platinum nanoparticles through free amino groups or cysteine residues in proteins, and, as a result, stabilize platinum nanoparticles by surface-bound proteins [53]. The PtNPs intracellular synthesis was found for non-pathogenic fungi *Neurospora crassa* [54]. *Saccharomyces boulardii* yeast can also produce intracellular PtNPs, where the amide and hydroxyl groups act as capping agents. [55].

Proteins as stabilizing agents for platinum nanoparticles was discovered by using yeast *Rhodotorula mucilaginosa* cell lysate supernatant for the PtNPs synthesis [50]. A very interesting study was made by Ito et al., who constructed yeasts *Saccharomyces cerevisiae*, displaying their hydrogenases from sulfate-reducing bacteria on the “cell membrane” and achieved Pt reduction in the nanoparticle form using engineered yeast [56]. Polysaccharides and proteins were found to play a role in the reduction and stabilization of platinum nanoparticles from *Cordyceps militaris* [57].

*Biosynthesis by cyanobacteria.* Cyanobacteria, a diverse group of photoautotrophic prokaryotes living in various ecosystems, can become an exciting bioreactor for platinum nanoparticle production. They are extremely curious in terms of PtNPs biosynthesis, because they are distinguished by their ability to fix atmospheric nitrogen (N_2_) by reducing nitrogen gas to ammonia using nitrogen reductase enzymes. Their potential for removing heavy metal ions from the environment gives them advantages in metal biotransformation [58,59]. Furthermore, they are full of different biomolecules, among them secondary metabolites, proteins, enzymes and pigments, possessing antimicrobial and antitumor activity [60]. Thus, PtNPs biosynthesis was found in cyanobacteria *Plectonema boryanum* by Lengke et al. [61]. The spherical Pt(II)-organic nanoparticles were connected into long bead-like chains by a continuous coating of organic material derived from the cyanobacterial cells and aged to nanoparticles of crystalline platinum metal with an increase in temperature and reaction time [61]. It is proposed that organic sulfur and phosphorus are involved in the reduction and complexation of platinum(IV) chloride [62]. In cyanobacteria *Anabaena*, *Calothrix*, and *Leptolyngbya*, PtNPs are synthesized intracellularly and then naturally released into the nutrient medium, where they are stabilized by polysaccharides, which allows them to be easily reduced [63]. Two different nitrogenases were determined in *Anabaena variabilis*, operating either under anaerobic or aerobic conditions in the heterocyst, while others can act only under aerobic conditions in the heterocyst and vegetative cells. At the same time, hydrogenase reduces the hydrogen ion to molecular hydrogen, that can be engaged in the reaction. So, nitrogenase and hydrogenase may play a crucial role in the PtNPs synthesis in cyanobacteria. Investigations of nanoparticle formation indicate that the intracellular nitrogenase enzyme is responsible for the metal reduction, but that the cellular environment is involved in the colloid growth process [64].

*Biosynthesis by algae*. Algae extracts contain a large number of biomolecules capable of reducing metal ions and capping them to improve their biocompatibility [65]. The antimicrobial and anticancer purposes of such compounds as alkaloids, flavonoids and terpenoids can be of great practical importance regarding algae use in platinum nanoparticle synthesis. Thus, PtNPs were received from the green algae *Botryococcus braunii* [66]. FTIR analysis data indicate proteins, polysaccharides, amides, and long chain fatty acids responsible for bioreduction and act as capping and stabilizing agents [66]. The platinum nanoparticle analysis of brown seaweed *Padina gymnosporia* revealed carbohydrates and proteins participation in the reduction of platinum ions (Pt) to PtNPs [67]. The red algae *Halymenia dilatata* aqueous extract contained alkaloids, flavonoids, tannins, terpenoids, steroids, carbohydrates, glycosides, amino acids, and protein, which can be attached by the functional groups to the surface of platinum nanoparticles and contribute to the metal ion’s reduction in the nanoparticle and stabilization by coating the surface [68].

*Biosynthesis by plants*. The popularity of plant extract-mediated nanoparticles have picked up speed in the last 30 years. Bioreduction involving mixing an aqueous plant extract with an aqueous solution of the corresponding metal salt can occur at room temperature and is usually completed within a few minutes [69]. Medicinal herbs carry a huge potential of biologically-active compounds contributing not only to the stabilization and synthesis of nanoparticles, but also increasing their biomedical significance. An important benefit of the plant-associated technique is the method of kinetics, significantly higher than other biosynthetic approaches equivalent to the chemical production of nanoparticles; additionally, plant physiology makes it possible to resist high concentrations of metals [10]. Reaction mixture staining in yellow or yellowish-brown color, depending on time, temperature, concentration, and pH, indicates the reduction of Pt^4+^ to Pt^0^ nanoparticles, and PtNPs formation [70]. The reduced solution is treated with ultrasound for some time to fractionate PtNPs from the biocompounds in the plant extract. Then, the solution is filtered by centrifugation and washed several times with distilled water to remove impurities. The produced PtNPs are dried and can be stored for further analysis [71].

Different plants are used for platinum ions reduction during PtNPs biosynthesis. Apparently, there is no completely universal pathway, and the synthesis depends on biomolecules contained in the extract of a particular plant. For example, in *Cacumen platycladi* extract, the key role belongs to reducing sugar (as part of saccharides) and flavonoids; in addition, an increase in temperature contributes to the process intensity [71,72]. In the case of PtNPs from *Punica granatum’s* peel extract, biosynthesis is associated with phenolic compounds, primarily with ellagic acid [73]. Pt^4+^ first chelated ellagic acid through its adjacent phenolic hydroxyls and formed an intermediate platinum complex. Due to the high oxidation–reduction potential of Pt^4+^, the adjacent phenolic hydroxyls of ellagic acid were inductively oxidized to the corresponding quinones. The Pt^4+^ was reduced to Pt^0^ in the presence of free electron or nascent hydrogen produced during the bio-reduction reaction [73]. The polyphenol involvement in the platinum salts was shown using *Barleria prionitis* leaf extract [74]. At the same time, the protein is a putative molecule responsible for the reduction of chloroplatinic ion into platinum nanoparticles in *Azadirachta indica* [75], while for *Diopyros kaki*, it is not an enzyme-mediated process [76]. Acid phosphatase performed an imperative role in the stability and capping of PtNPs from *Rumex dentatus* seeds extract [77]. Based on the data on PtNPs synthesis using gum kondagogu (*Cochlospermum gossypium*), the authors [78] suggested that nanoparticles are formed on the gum surface and not in solution. The first stage involves the capture of metal ions on the gum network surface, possibly through electrostatic interaction between metal ions and negatively-charged carboxylate and hydroxyl groups in the biopolymer. Sugars, amino acids and fatty acids can be both reducing and capping agents in the platinum nanoparticle formation in Gum kondagogu. Upon hydrolysis, metal ions lead to the formation of metal nuclei, and they subsequently grow and accumulate in the form of nanoparticles within the gum matrix [78].

Capping agents have attracted special attention from researchers because they can make a special contribution to the structure of platinum nanoparticles and improve their biomedical properties, possessing their own. These exciting compounds can take part both in the reduction of platinum ions to Pt^0^ and stabilization against agglomeration [23,79,80,81]. Flavonoids, tannins, glycosides, alkaloids, steroids, terpenoids, saponins, polysaccharides, proteins and enzymes can be represented as such agents (Table 1) [35,73,74,78,82,83,84,85,86,87,88,89,90,91,92,93,94,95,96,97,98,99,100,101,102,103]. The bio reduction pathway is heavily influenced by functional groups such as amine (–NH), alcohol (–OH), and carboxylic group (–COO). Groups such as hydroxyl are oxidized during the reduction and stabilization process, resulting in oxidized forms that start capping the surface of the PtNPs [87]. Additionally, the carbonyl, carboxylate, and amine groups normally bind to the surface of the NPs and keep away from their aggregation as well as stabilize the NPs [97,98].

*Biosynthesis by other bio-objects*. More exotic biological objects were found to be used in PtNPs synthesis. So, platinum nanoparticles were synthesized using sheep milk by Gholami-Shabani et al. [104]. Milk proteins bind to PtNPs through the free anime group and may be responsible for the reduction in platinum ions; they also capped the surface of the nanoparticles and kept them stable for longer periods [104]. Honey-mediated platinum nanoparticles is another specific variant, where honey is responsible for the nanoparticle’s reduction and stabilization using the electrostatic interaction of carboxylate in groups of amino acid residues in protein with platinum [105]. PtNPs were also produced enzymatically: acid phosphatase isolated from seeds of *Cichorium intybus* effectively synthesized platinum nanoparticles [77].

Thus, PtNPs biosynthesis opens up wide horizons for nanoparticle production with a wide diversity of properties and opportunities for their use in biomedical purposes (Figure 3).

## 3. Application of Green PtNPs

### 3.1. Antibacterial Activity

The large-scale application of antibiotics to treat infectious disease has led to the emergence of a large number of pathogen strains resistant to antibiotics causing huge problems in medicine. Biofilms formed by pathogenic microorganisms are a separate issue. Metal nanoparticles are of great value in the solution to this problem, being an effective basis for the development of antimicrobial drugs. The bactericidal properties of metals such as gold and silver have been known to mankind for a long time, and modern methods of nanotechnology discover new possibilities for their use [106,107]. Metal nanoparticles, including PtNPs, exhibit remarkable biocidal properties against both gram-positive and gram-negative bacteria [108]. Such antimicrobial activity depends on the surface area in contact with microorganisms; their small size and high surface-to-volume ratio gives them the opportunity to interact closely with microbial membranes [107,109]. The nanoparticle shape also plays an important role in antibacterial activity, and capping agents, due to their own antimicrobial properties, are able to enhance it [110].

The mechanisms of nanoparticle action on bacterial cells include: destruction of the microbial cell wall and cell membrane, pump mechanisms damage; destruction of cellular components: ribosome breakdown, inhibition of deoxyribonucleic acid (DNA) replication and enzyme dysfunction; formation of reactive oxygen species (ROS) and induction of oxidative stress; and triggering of both congenital and adaptive host immune responses and inhibition of biofilm formation (Figure 4) [20,111]. Metallic nanoparticles interact with the bacterial cell wall by attraction between the negative charge of the cell wall and the PtNPs positive charge [20,112]. The main functions of the bacterial cell wall and cell membrane are protection from external influences and the transport of nutrients into and out of the cell. Both Gram-negative and Gram-positive bacteria surface are negatively charged due to the presence of lipoteichoic acid (in Gram-positive) and lipopolysaccharide (in Gram-negatives) [113]. The outer membrane of Gram-negative bacteria is a lipid bilayer, whereas the inner membrane is composed of phospholipids. One of the major differences between Gram-positive and Gram-negative is a thicker peptidoglycan layer in the cell wall of Gram-positive, which makes them less vulnerable to metal nanoparticles [114]. In addition, the greater nanoparticle efficiency to Gram-negative may be the result of the presence of lipopolysaccharides (LPS) carrying a negative charge, which ensures the PtNPs adhesion to the bacterial cell wall. Due to the presence of the rigid cell wall of Gram-positive bacteria, the antimicrobial activity of Gram-negative bacteria was higher compared to Gram-positive bacteria. Therefore, the biogenic nanoparticles carrying amino groups on their surfaces can attach more effectively to both Gram-negative and Gram-positive bacterial cell walls and destroy them [50]. As a result of interaction between PtNPS and the bacterial cell wall, morphological and permeability function changes in the membrane is observed, the integrity of the bacterial cell is disrupted and death occurs [20]. In case of PtNPs, their strong negative zeta potential enhances antibacterial activity [110].

Penetrating into the cell, platinum nanoparticles complete the process: binding to DNA, they transfer it from the normal state to the condensed one, leading to the loss of replication ability; binding to thiol, phosphorous or sulfhydryl (-SH), amine and carboxylic groups of enzymes results in their inactivation [101]. The loss of hydrogen ions in proteins leads to the cell membrane destruction, increasing its permeability to PtNPs, ultimately inhibiting bacterial metabolism and initiating cell death [77]. The antibacterial activity of PtNPs is associated with ROS (reactive oxygen species) generation: they increase ROS number in bacterial cells. ROS include highly reactive radicals −OH, H_2_O_2_ and less toxic radicals O^2−^, able to affect DNA, RNA and proteins, causing the death of bacteria [77]. These ROS interact directly with PtNPs, causing protein degradation and lipid breakdown, resulting in the down-regulation of DNA, oxidative stress, and finally apoptosis of bacterial cells [77,115]. An important role in maintaining the intracellular redox environment is played by the reduced glutathione (GSH), a non-protein tripeptide that protects cells from oxidative damage by absorbing ROS. However, excessive ROS formation can oxidize GSH to glutathione disulfide. The decrease in the concentration of GSH in cells treated with platinum nanoparticles was determined [116]. Interaction PtNPs with the 30S ribosome subunit induces protein synthesis to stop [117]. Some studies suggested that bacterial growth inhibition is interconnected with ATP production and mitochondrial membrane potential [118]. The effect of platinum nanoparticles on DNA synthesis was shown by PtNPs ability to inhibit the Taq DNA polymerase and affect the secondary structure of DNA in higher concentrations [119]; a one-hour treatment of *Salmonella enteritidis* with PtNPs removed part of the DNA from the bacterial cell [120]. The influence on the proposed mechanism of platinum nanoparticles on the bacterial cell is presented in Figure 4. The size of platinum nanoparticles is of great importance in antibacterial activity: small particle sizes often have a higher surface area and so are more effective than those with larger particle sizes; for example, a smaller size PtNPs could pass more effectively through the thick cell wall of Gram-positive bacteria [50].

All of the above is confirmed by the influence of platinum nanoparticles on different bacterium types. Thus, an inhibitory effect is shown for gram-negative bacteria *Pseudomonas aeruginosa* [50], *S. typhimurium* [100], *Klebsiella oxytoca* and *Klebsiella aerogenes* [101], *S. typhi* [116], *Ps. aeruginosa* [118], *Enterobacter aerogenes*, *S. enteritidis* [120], *Klebsiella pneumoniae* [84,121], *B. licheniformis* [122], *E. coli* [77,123], *Aeromonas hydrophila* [63,124], *Proteus vulgaris* [121], *Enterobacter cloacae* [121]; Gram-positive bacterium—*Staphylococcus aureus* [50], *Listeria innocua* [50], *Streptococcus pneumonia* [68], *Lactococcus lactis*, *Bacillus subtilis* [89], *St. epidermis* [101], *St. haemolyticus* [125], *Enterococcus faecium* [3,125], *St. pyogens* and *Vibrio cholerae* [126].

### 3.2. Anti-Fungal Activity

There are numerous antifungal drugs, but most of them have many side effects. Green PtNPs represent an alternative in solving this problem. The received data allow us to characterize them as an effective antifungal agent. Anti-fungal activity was found for *Fusarium oxysporum* [66], and also plant pathogenic fungi such as *Colletotrichum acutatum*, and *Cladosporium fulvum* [98]. PtNPs prepared using *X. strumarium* leaves extracts showed significant anti-fungal activity against *Candida albicans*, *C. tropicalis*, C*. parapsilosis*, *Aspergillus flaves* and *A. niger* [91]. The fungicidal effect of platinum nanoparticles obtained with gum kondagogu was found for *A. parasiticus* and *A. flavus* [127]. The inhibitory activity of Pt nanocomposite on *A. parasiticus* and *A. flavusfungi* is mediated through the induction of oxidative stress, resulting in the formation of ROS and subsequent damage to fungal mycelial morphology and membrane, ultimately leading to cellular damage and fungal growth [127].

### 3.3. Anti-Cancer Activity

Cancer is one of the biggest threats to humanity, occupying second place in terms of mortality. Due to the lack of high-quality diagnosis and treatment in the early disease stages, cancer mortality rates are significantly higher in low- and middle-income than in developed countries. Moreover, most modern drugs targeted at treating cancer have a huge number of adverse effects that can seriously worsen the quality of a patient’s life. Therefore, searching for new, inexpensive drugs possessing a low toxic effect would allow this problem to be solved on a qualitatively new level.

Platinum–based anticancer drugs have been known for a long time. One of these is cisplatin, with strong cytotoxic, bactericidal and mutagenic properties. The anticancer activity of cisplatin was discovered back in 1965 by Barnett Rosenberg [128]. Its action is based on the ability to form strong specific bonds with DNA that induce chemical damage to DNA bases. However, there are many negative effects of such drugs, including nephrotoxicity, fatigue, emesis, alopecia, peripheral neuropathy, and myelosupression. Platinum nanoparticles can open a new page for cancer treatment. The most promising are PtNPs mediated by plant extracts containing essential oils, acids, alkaloids, phytoncides, which show efficacy in various types of cancers [129]. For example, an anti-cancer effect was found for *Aloe vera, Catharanthus roseus*, hot pepper (*Capsicum annuum*), tulasi (*Ocimum sanctum*) and many other medicinal plants [130]. The combined effect of platinum nanoparticles, capping by different herbs compounds, could enhance the anti-cancer effect and eliminate the toxic effect of the metal.

To create new therapeutic agents against cancer, it is very significant to develop inducers of apoptosis: programmed cell death coordinated by a cascade of interdependent cellular reactions. Moreover, it is the most important process for maintaining homeostasis between cell proliferation and mortality [131].

Although the action mechanism of platinum nanoparticles on cancer cells is not fully studied, the basic principles, based on in vitro experiments, can be selected: (a) cell cycle arrest; (b) penetration into the nuclear, nucleus and DNA fragmentation; (c) the level of glutathione; (d) inducing mitochondrial dysfunction; (e) expression of caspases; (f) expression of different proinflammatory cytokines; (g) increasing the level of different enzymes (superoxide dismutase, lactate dehydrogenase etc.); and (h) ROS generation (Figure 5).

Apoptosis induction by platinum nanoparticles was determined through G0/G1 cell cycle arrest [32]. The influence of PtNPs on cell cycle progression was shown [16]. A significant increase in the percentage of cells in the sub-G1 phase was found after treatment with nanoparticles, and the G0/G1 phase was shown to decrease along with the increase in the sub-G1 phase [16]. Apoptosis was noted in the part of the G0/G1 region, indicating that the G1 phase cells were lost by programmed cell death [32]. PtNPs-treated cells showed higher cell growth at the G2/M phase, which revealed that in the G2/M phase, induced cell cycle was arrested and cell numbers were increased in the sub G0 cell death phase, which revealed the application of PtNPs for the treatment of cervical cancer treatment [31]. Platinum nanoparticles also induced changes in cell morphology: for instance, the appearance of long cytoplasmic protrusions was determined in SH-SY5Y cells, a cloned subline of a neuroblastoma cell line [132], and density; membrane disintegration increased cell population in apoptosis and caused chromatin condensation and nucleus fragmentation [16,103,133]. The mechanism of anticancer activity was found to be an externalization of phosphatidyl serine and an increase in membrane permeability, which are considered to be the hallmarks of apoptosis [74]. Additionally, in A549 cells, the proliferative activity decreased gradually over time in proportion to the increasing concentration of the platinum nanoparticles [102].

The literature data reveal that the antitumor activity of PtNPs may be associated with penetration into the nucleus and mitochondria of the cell, inhibition of DNA replication and mitosis by binding to a DNA molecule, leading to apoptosis [134]. The PtNPs inhibited the DNA replication and affected the secondary structure of DNA at higher concentrations [114]. Mitochondria are responsible for energy production and are also necessary for the primary pathway regulation of the apoptosis and non-apoptotic cell death [135]. The mitochondrial membrane potential (MMP) loss evokes disturbances in the ATP synthesis, leading to ATP accumulation in the mitochondria. Exposed to platinum nanoparticles, low levels of MMP and ATP indicated mitochondrial dysfunction. Higher expression of pro-apoptotic genes and lower expression of anti-apoptotic genes were also observed by the influence of PtNPs [136]. Inhibition of mitochondrial respiration was found in HeLa cells, and the polarization of mitochondria under the action of PtNPs induced a large loss in ATP and mitochondrial dysfunction and led to apoptosis in human cervical cancer cells [95]. PtNPs decreased the level of MMP in various types of cancer cells, including human neuroblastoma cancer cells SH-SY5Y, human monocytic THP-1 cells, and human bone osteosarcoma epithelial cells [132].

PtNPs increased the expression of caspases, playing a substantial role in cell apoptosis [137]. The downregulation of p53 (protein p53) may also be a key element of anticancer activity, because it is a transcription factor regulating the cell cycle and acting as a suppressor of malignant tumor formation [138]. p53 also induces the expression of pro-apoptotic Bcl-2 family members such as Bax, Puma, and Noxa in response to the mitochondrial apoptosis pathway activation. On the surface of mitochondria, these pro-apoptotic proteins meet with anti-apoptotic members of the Bcl-2 family. DNA damage is determined by the ratio of pro-apoptotic and anti-apoptotic proteins. Furthermore, pro-apoptotic signals result in the release of cytochrome C from mitochondria, followed by the activation of cysteine caspases [139]. Bcl-2 protein plays an essential role in the apoptosis process, which activates caspase-9 and caspase-3, triggering the apoptosis cascade (with the participation of another caspases-7,8) [140]. Almeer et al. suggested that PtNPs synthesized using leaf extract of *Azadirachta indica* initiate an internal apoptosis pathway, mediated by an increase and decrease in the expression of Bax and bcl-2 in mitochondria [137]. PtNPs exposure in vitro leads to p53activation in cells caused by genotoxic stress, followed by activation of p21, leading to a stop in the growth of proliferating cells in the S-phase and subsequent apoptosis [119].

Cellular redox homeostasis change in a significant part of the signaling pathway due to the overproduction of intracellular ROS puts the cell on the apoptosis pathway [141]. Moreover, ROS-mediated transcription factors control the expression of various genes involved in inflammation, cell transformation, tumor cells death or survival, proliferation, invasion, angiogenesis and metastasis. Platinum nanoparticles from *T. involucrata* are able to enhance cellular apoptosis due to ROS over-production [95]. Similar data were obtained for HEK293 (human embryonic kidney), MCF-7 (human breast cancer) and HepG2 (hepatocellular carcinoma) cells, where platinum nanoparticles induced cytotoxicity and apoptosis via the generation of ROS [137,142].

Possessing antioxidant properties, GSH not only protects the cell from toxic free radicals, but also generally determines the redox characteristics of the intracellular environment. ROS generation was discovered to convert GSH to GSSG (Glutathione disulfide) through the oxidation process [143]. Oxidized glutathione is reduced by the enzyme glutathione reductase induced by oxidative stress. The most important parameters showing the oxidative stress level are the ratio of reduced and oxidized glutathione forms in the cell. ROS generation in a concentration and time-dependent manner, and a decrease in GSH levels in HEK293 (human embryonic kidney) cells treated with platinum nanoparticles led to damage to the cellular component [137]. PtNPs can decrease the various antioxidant levels in different types of cancer cells, initiating their apoptosis [132].

Additionally, nanoparticles also promoted the expression of different proinflammatory proteins: cytokines such as tumor necrosis factor TNF-α, TGF-β, and NF-κB, and interleukin-1β (IL-1β), IL-6, IL-8 [32,133,144]. Platinum nanoparticles enhance the lactate dehydrogenase level and increase apoptosis and oxidative DNA damage [145], as well as superoxide dismutase activity (SOD), lipid peroxide (LPO) and malondialdehyde (MDA) levels [137,146]. Lactate dehydrogenase is a characteristic cell death marker, released into the surrounding extracellular space when cell membranes are disrupted [132]. LPO arises from the oxidation of fatty acids induced by oxidants; therefore, it is also a characteristic sign of negative cellular effects. The production of LPO-derived aldehydes in cancer cells depends on the presence of ROS. The increased ROS level can increase the formation of LPO products and eventually increase oxidative damage to DNA [147].

Abnormally increased production of Nitric Oxide (NO) triggers cell damage [148]. An increased NO level results in mitochondria damage through a change in membrane potential and inhibition of respiratory chain, and also induces a sequence of events in the cell, leading to the ROS generation, loss of mitochondrial membrane potential, the release of cytochrome C into the cytosol, activation of caspases, DNA fragmentation and, ultimately, apoptosis [148,149,150]. PtNPs in combination with retinoic acid have the ability to enhance NO production in SH-SY5Y cells and cause cancer cell death [132].

Apoptosis stimulation is possible by the induction of ERS (Endoplasmic reticulum stress). As a result, an unfolded protein response (UPR) is initiated to restore cellular homeostasis or induce apoptosis [151]. UPR is regulated by various transmembrane proteins, such as protein kinase-like ER kinase (PARK), inositol-requiring enzyme 1 (IRE1), activating transcription factor (ATF6), andATF4, involved in maintaining homeostasis. The inducing apoptosis through the induction of ERS was shown to be possible via platinum nanoparticles [132].

Various research display anticancer activity in vitro against HeLa cells [85,91,152], HEK293 cells [137,153], MCF-7 (breast cancer) and HepG-2 (hepatocellular carcinoma) [32,49,103,104,142], MDA-MB-231 breast cancer cell line [154], lung carcinoma cells A549 [102,131], SH-SY5Y cells [132], ovarian teratocarcinoma (PA-1), pancreatic cancer(Mia-Pa-Ca-2) cells [16], colon carcinoma cells (HCT-116) [90], sarcoma-180 (S-180) cells [155], 4T1 breast cancer cells [156], A431 cell lines (epidermoid carcinoma) [55] and myoblast C2C12 carcinoma cells [144].

The anticancer activity data are not limited by in vitro studies. Thus, in vivo experiments showed PtNPs at the medium and high doses effectively inhibited and delayed the growth of lung cancer in severe combined immune deficient mice [157]. The effect of platinum nanoparticles on breast cancer cells was found for PtNPs that contribute to breast cancer metastasis by damaging the vascular endothelial barrier. PtNPs disrupt the proliferation and migration of endothelial cells and the formation of tubular structures, destroy endothelial junction adhesions and induce the impermeability of the endothelial barrier in vitro. It is assumed that such a stimulation occurs through ROS generation, changing in the expression and conformation of endothelial connective proteins, thereby contributing to intravasation and extravasation of implanted breast cancer cells 4T1 and leading to cancer metastasis in female mice BALB/c nude in vivo [156].

### 3.4. Antioxidant Activity

Reactive oxygen species such as hydroxyl, epoxyl, superoxide, peroxylnitrile, and singlet oxygen generate oxidative stress, leading to the growth of various diseases such as inflammation, atherosclerosis, aging, cancer, and neurodegenerative disorders [158]. Free radical activity suppression by antioxidants helps to support the immune system and allows it to fight against viruses and other foreign invaders more effectively. An evaluation of the antioxidant properties of PtNPs in vitro, as a rule, is performed by removing DPPH (2,2-diphenyl-1-picrylhydrazyl) radicals, as one of the most important and widespread free radicals that can harm human cells [159]. DPPH is an uncharged free radical that can accept hydrogen or free electrons to produce a stable diamagnetic molecule, that is why it has long been used to test the free radical scavenging capacity of antioxidants [159,160]. The antioxidant activity is reflected as a percentage in the DPF absorption or removal [159,160]. It was shown that antioxidant activity was found to be dose-dependent for PtNPs and may also correlate with the nanoparticle size and the zeta potential, depending on the used “bio-factory” [50,66,68,96,123]. For example, the smallest nanoparticles produced by gram-negative bacteria showed better antioxidant activity than gram-positive ones [50]. In addition, capping agents also seemed to play quite an important role in the PtNPs antioxidant activity [77]. It should be noted that platinum nanoparticles synthesized by plants with antioxidant potential are of great importance, because plant compounds can prevent ROS-triggered oxidative damage when the endogenous antioxidant system does not cope independently. Ascorbic acid can be an example, possessing an important antioxidant value in physiological conditions and pro-oxidant in pathological conditions (bacterial infections or cancer) [161]. Antioxidant activity data were found for plant-mediated platinum nanoparticles from *T. involucrata* [95], *A. halimus* [89], *D. bulbifera* [86], *Salix tetraspeama* [87] and *Cordyceps militaris* [100]. Antioxidant and neuroprotective activities studies of platinum nanoparticles synthesized by *Bacopa monnieri* leaf extract showed a decrease in ROS generation and the free radical’s removal, thereby increasing the levels of dopamine, its metabolites, GSH, catalase, SOD, and complex I, and decreasing MDA levels along with enhanced motor activity with MPTP-induced (1-methyl 4-phenyl 1,2,3,6 tetrahydropyridine) neurotoxicity in a model of Parkinson’s disease in Danio fish [85]. The obtained data may become a potential option for the fight against Parkinson’s disease. Suppression of reactive oxygen species by means of PtNPs interacting with antioxidant enzymes such as superoxide dismutase, catalase and glutathione peroxidase was shown using *Drosophila melanogaster* as an in vivo model system. Moreover, platinum nanoparticles interacted with hemocytes without any toxic cell effect and significantly accelerated the wound healing process in a short time [162].

### 3.5. Anti-Diabetic Activity

Diabetes mellitus (diabetes) is one of the most urgent problems faced by mankind, despite the significant amount of well-established diagnosis and treatment methods for this disease. Alpha-amylase and α-glucosidase are known to be key enzymes in carbohydrate metabolism, so their inhibition is one of the most significant strategies for diabetes therapy [163,164]. In addition, α-glucosidase is also considered as the main enzyme involved in carbohydrate metabolism, catalyzing the cleavage of oligosaccharides and disaccharides into monosaccharides [163,164]. An amylase inhibitor, jointly with starchy foods, reduces the usual upturn in blood sugar. Amylase inhibitors or starch blockers, including silver and other metal nanoparticles, were proved to prevent the absorption of food starches by organism [163]. The anti-diabetic effect was discovered in vitro for PtNPs from *P. salicifolium*: a mild inhibitory effect against α-amylase and a higher effect against α-glucosidase was exhibited [96]. The authors suggested that the antidiabetic effect could be attributed to the direct correlation between the phytoconstituents surrounding PtNPs and α-amylase and/or α- glucosidase inhibitory action, excessive ROS production or imbalanced antioxidant protection mechanisms [96]. Furthermore, a significant decrease in glucose levels after PtNPs injection was observed in streptozotocin-induced diabetic rats [165]. Type 2 Diabetes is characterized by an increase in ROS production level, induced by chronic extracellular hyperglycemia as a result of a violation of the cell redox state, causing the abnormal expression of insulin sensitivity genes [166,167]. In this regard, the enzyme-like antioxidant properties of platinum nanoparticles to absorb free radicals and decline ROS concentration can promote the diabetes struggle. For instance, under the influence of PtNPs, the induction of the gene expression of the antioxidant enzyme catalase (CAT), glutathione peroxidase (GPx) and hemoxoigenase, suppression of fasting blood glucose levels and an improvement in the impaired ability to sugar tolerance in obese insulin-resistant type 2 diabetic KK-Ay mice was shown [168].

### 3.6. Anti-Inflammation Activity

ROS overproduction is associated with the pathogenesis of inflammatory diseases. Antioxidant therapy to solve this problem is possible in the face of platinum nanoparticles. Rehman et al. demonstrated that in vitro anti-inflammatory activity of PtNPs may be attributed to their down regulation of the NFjB signaling pathway in macrophages in lipopolysaccharide-stimulated RAW 264.7 cells [169]. PtNPs showed direct anti-inflammatory activity in RAW264.7 macrophages through a mechanism involving the intracellular ROS uptake by suppressing lipopolysaccharide-triggered production of proinflammatory mediators, including nitric oxide, tumor necrosis factor-α and interleukin-6 [170]. The high antioxidant activity of platinum nanoparticles was found in a cavernous malformation cellular model of the human brain [171]. PtNPs had a significant neuroprotective effect on the ischemic mouse brain [172] and effectively protected keratinocytes from UV-induced inflammation [173] and suppressed chronic obstructive lung inflammation provoked by acute cigarette smoking [174]. Platinum nanoparticles, alone and in combination with palladium nanoparticles, showed antioxidant activity and weakened aging-related skin pathologies in vivo in mice, without causing morphological abnormalities such as cellular infiltration, fat deposition, or cell damage in mouse skin [175]. It was supposed that the catalase activity shown by the nanoparticle combination may be useful in the treatment of vertigo: an acquired pigmentation disorder characterized by H_2_O_2_/peroxynitrite-mediated oxidative and nitrative stress in the skin.

### 3.7. Other Application

*Photothermal therapy and radiotherapy*. Antitumor chemotherapy has many negative diverse consequences for patients, therefore developing less harmful and cancer-specific strategies is an extremely important task. Photothermal therapy may be the one of the decisions. This non-invasive treatment assumes that PtNPs increase the cellular temperature upon irradiation, causing DNA/RNA damage, membrane rupture, protein denaturation and finally apoptosis [176,177]. Such photothermal therapy using PtNPs, 5–6 nm in size, induce damage of a selective cellular component and cell death [38,176]. Polypyrrole-coated iron-platinum nanoparticles were used for photothermal therapy and photoacoustic imaging. In vitro investigation experimentally demonstrated the effectiveness of these NPs in killing cancer cells with NIR laser irradiation. Moreover, the phantom test of PAI used in conjunction with FePt@PPyNPs showed a strong photoacoustic signal [177]. Cysteine surface modified FePtNPs can be potential sensitizers for chemoradiotherapy: in vitro NPS FePt-Cys induced ROS, suppressed the antioxidant protein expression and induced cell apoptosis, and also facilitated the chemoradiotherapy effects by activating the caspase system and disrupting DNA damage repair. The drug safety and the synergistic effect with cisplatin and irradiation was confirmed by in vivo studies [178]. FePtNPs could potentially be a new strategy for increasing radiation therapy efficiency in cancer cells overexpressing hCtr1 due to enhanced uptake and targeting of mitochondria [179]. The synergistic antitumor effect with radiation to eliminate tumors for MFP-FePt-GONCs nanoparticles was determined [180]. These nanoparticles improved the radiation effects by activating internal mitochondrial-mediated apoptosis and worsening DNA damage repair. Additionally, they induced ROS release, which suppressed antioxidant protein expression and induced cell apoptosis [180]. A combination of PtNPs with irradiation by fast ions effectively enhances the strong, lethal damage to DNA [181]. In spite of the data presented above, which were obtained for platinum nanoparticles/nanocomposites synthesized by physicochemical methods, “green” PtNPs may have serious potential in this method. Platinum nanoparticles data obtained by *Prosopis farcta* fruit extract indicate their stability and biocompatibility for application as a contrast agent in computed tomography as an alternative to low molecular weight agents with toxic effects [153].

*Catalytic activity*. The excellent catalytic activity of “green” PtNPs was shown for removing pharmaceutical products (PhP). The platinum nanoparticles produced via *D. vulgaris* worked as an effective biocatalyst in the removal of four PHPs classes: ibuprofen, ciprofloxacin, sulfamethoxazole and 17β-estradiol, which are most relevant in the environment [182]. It is important to note that only 13% of catalytic activity was lost during recycling, indicating the possibility of bacteriogenic PtNPs reuse for technological development in the pharmaceutical wastewater treatment [182].

*Detection*. Platinum nanoparticles obtained by physicochemical approaches can be used for the detection of DNA, cancer cells, antibiotics, glucose, proteins, bacteria, viruses and antibodies [38,183,184,185,186,187,188,189]. The peroxidase activity of plant-mediated PtNPs makes it possible to quickly detect Hg ions [190], as well as hydrogen peroxide [82]. Additionally, PtNPs were successfully used for hydrazine detection in spiked water samples [82].

## 4. Toxicology of «Green» Platinum Nanoparticles

The numerous positive properties of PtNPs, such as antibacterial, anticancer and others, however, do not cancel their toxicity research for both healthy and pathogenic cells. The toxicity determination of platinum nanoparticles is a mandatory procedure before their introduction into medical practice. Cytotoxicity against cancer cells has already been confirmed by the numerous studies mentioned above. Furthermore, for example, PtNPs entered to chicken embryos at the beginning of embryogenesis in concentrations from 1 to 20 micrograms/mL did not affect the embryo growth and development. Although the neurotoxicity study after the influence of PtNPs did not reveal changes in the cell number in the cerebral cortex, the ultrastructure analysis of brain tissue revealed mitochondria degradation, apoptosis activation, as well as a decrease in the rate of brain cells proliferation, which could potentially be used for brain cancer therapy [191]. PtNPs induced hatching delays, as well as a concentration-dependent drop in heart rate, touch response and axis curvatures in *Danio rerio* embryo [192]. PtNPs had acute toxic effects on cardiac electrophysiology and can induce threatening cardiac conduction block. These acute electrophysiological toxicities of PtNPs are most likely caused by a nanoscale interference of PtNPs on ion channels at the extracellular side, rather than by oxidative damage or other slower biological processes [193]. At the same time, the presence of platinum nanoparticles in monocytes effectively reduced ROS generation, which did not affect cell viability, and the expression of cytokines and chemokines were not disturbed; therefore, PtNPs were non-toxic and biocompatible [194]. PtNPs application at concentrations below 20 micrograms/mL did not lead to any measurable cytotoxicity in the cell line of non-malignant human embryonic kidney HEK293, but at higher concentrations PtNPs were cytotoxic and diminished cell viability [195]. PtNPs showed cytotoxicity against normal cells, changing the morphology and density of cells, causing chromosome condensation, blocking cell proliferation and inducing apoptosis [196]. However, the acute cytotoxicity absence of PtNPs in vitro for human renal tubular epithelial cells (HRTECs), human keratinocytes (HaCat), human dermal fibroblasts (HDFs), human epithelial kidney cells (HEK 293), and primary human coronary artery endothelial cells (HCAECs) was observed [197]. PtNPs described above were produced by physical and chemical methods; however, a toxic effect was not shown for “green synthesis”. An in vivo toxicity study was performed in *Artemia* as one of the ideal crustacea for testing acute toxicity in laboratory conditions. *H. dilatata* aqueous extract-mediated PtNPs do not induce major mortality in *Artemia nauplii* [68]. No significant cytotoxic effect in vitro was observed in the normal PBMC (Peripheral blood mononuclear cells) at the highest concentration (200 μg/mL), but ptNPs induced cell death in ovarian, lung and pancreatic cancer cell lines; additionally, the nanoparticle cytotoxicity was dependent on the cell type [16]. Antitumor activity was determined for *Curvularia*-based platinum nanoparticles, but with subchronic exposure (3 months) they did not have significant toxicological manifestations of renal and hepatic tissue, and also decreased oxidative stress and improved liver function in mice with tumors compared with untreated control groups [155]. *Pr. farcta* fruit extract-associated PtNPs were nontoxic at relatively high concentrations (100 μg/mL) [153]. It is extremely interesting that PtNps with protein coronas were formed in the blood of humans treated with cisplatin [198]. Amino acids, peptides, and proteins were assumed to be capping agents. These self-assembling PtNPTs were rapidly formed and accumulated in tumors. The size of these PtNPs ensures a long half-life with slow penetration through the glomerular filtration barrier. They were found as safe to use and can act as antitumor agents to inhibit the growth of a tumor resistant to chemotherapy by consuming intracellular glutathione and activating apoptosis [198]. In addition, the long-term toxicity estimation of the drug did not show pathological changes in the mouse organs, weight loss, and immune response cells [198].

## 5. Conclusions

“Green” platinum nanoparticles are of particular interest in solving various medical problems. Biological systems capable of becoming a biofactory for PtNPs production are full of useful compounds and can also have pharmaceutical applications, coupled with nanoparticles. Apparently, plants are the most interesting because they have powerful therapeutic potential themselves, contributing not only to PtNPs production and increasing their medicinal properties, but also mitigating the negative, including cytotoxic, effects on healthy cells. On the other hand, although microbial synthesis of PtNPs is not so popular due to the complexity of selecting cultivation conditions, its capabilities also seem promising, allowing a target object using genetically engineered microorganism strains to be able to produce the necessary bio-compounds for PtNPs synthesis. The special place in platinum nanoparticles synthesis should be given to compounds directly involved in the platinum ion reduction, and especially capping agents, playing a crucial role in the influence of PtNPs on living objects. These implicated biomolecules are already known for plant-mediated platinum nanoparticles [78,79,80,81,82], however, unfortunately, little data about reducing and capping agents for bacterial and fungal PtNPs were found.

The experience of studying the biological properties of platinum nanoparticles produced by physicochemical methods can be transferred to biologically-obtained PtNPs. For example, the inhibitory effect of PtNPs stabilized by polyacrylic acid on oral bacterium *Str. mutans* biofilm formation was observed [199]. An important role in this process is played by glucosyltransferases, metabolizing sucrose into glucose with the cariesogenic biofilm formation. In this case, metal nanoparticles are capable of imitating a protein, having a metal core and a surface covered with organic biopolymers [200]. Such a protein-like structure helps to inhibit biofilm formation. The antibacterial activity of “green” platinum nanoparticles has been described in many studies, but anti-biofilm formation is poorly highlighted in literature. The suppression of biofilm formation in *S. typhy* was found [116]. Such data contribute to the fight against pathogenic microorganisms, especially antibiotic resistance. For example, it is known that platinum nanoparticles can adsorb the homologue of human a2-macroglobulin, which is produced by *Ps. aeruginosa*, in order to use host cells to stimulate their survival [201,202]. This interaction with PtNPs can have an antibacterial effect.

Special attention is paid to the anti-cancer properties of PtNPs. Along with the already known platinum-based drugs, PtNPs appear to be promising objects, both for direct treatment and for drug delivery to diseased organs. The action mechanism on cancer cells involves triggering various signals inside the cell, eventually leading to its death. In most experiments, a dose-dependent effect was shown: an increase in the concentration of PtNPs leads to the level increase in lactate dehydrogenase, ROS, malondialdehyde, nitric oxide, and carbonylated protein, while at the same time a significant decrease in the levels of reduced glutathione, thioredoxin, superoxide dismutase and catalase against the background of oxidative stress [132,145,203]. The mechanism of cell death was confirmed by mitochondrial dysfunction and decrease in ATP levels. In addition, cell apoptosis was observed through G0/G1 cell cycle arrest [16,32]. The platinum nanoparticles with anticancer activity were synthesized not only via plant extracts capable of carrying biomolecules with their own anti-cancer effect, but also by specific plant compounds; for example, apigenin, a natural antioxidant with anti-inflammatory and anti-carcinogenic properties [204], tangeretin, a flavone contained in tangerine peel and others citrus fruits that strengthen the cell wall and acts as a protective plant mechanism against pathogens [145] and the carotenoid pigment lycopene [205]. In the latter case, PtNPs can change the gene expression involved in protein misfolding, mitochondrial function, protein synthesis, inflammatory reactions, and transcription regulation [205]. PtNPs were found to stimulate exosome biogenesis, induce oxidative stress, and ceramide signaling pathway [206]. An important factor in favor of platinum nanoparticles produced through biological methods is the genotoxic and proapoptotic effects for cancer cells similar to the cisplatin effect [136], while in most studies with normal cells no negative effects were found [207]. For example, PtNPs evoke toxic effects on primary keratinocytes, decreasing cellular metabolism, but these changes do not affect cell viability or migration. In addition, the size effect of the nanoparticles was also determined: smaller NPS had a more detrimental effect on DNA stability than larger ones; the change in the caspases 9 and 3 activity was also induced, mainly by smaller PtNPS [208]. Moreover, apparently PtNPs can have selective cytotoxicity towards different types of cancer cells [209]. Platinum nanoparticles produced via physical approach were able to overcome the resistance of some tumors to radiation, representing a potential major breakthrough in radiation therapy [210]. Such studies for “green” platinum nanoparticles, coupled with their biocompatibility, could be extremely useful in the struggle against cancer. Nevertheless, it is necessary to develop strategies taking into account the PtNPs capacity to penetrate biological barriers and to reach the therapeutic site in the necessary doses with minimizing accumulation in undesirable places [211], because, for example, 30–99% of injected nanoparticles will accumulate and isolate in the liver after administration into the body. This results in the delivery decrease to the affected target tissue and potentially leads to increased toxicity at the cellular level in the liver [212].

Other properties of platinum nanoparticles: antioxidant, anti-diabetic, catalytic, drug delivery, are also very attractive for practical application [213]. Biosynthesis may be a more effective approach for obtaining new tools for future therapeutic methods of drug delivery in the treatment of cancer, inflammation, diabetes, and possibly antiviral therapy. A specific molecule’s conjugation with the surface of nanoparticles can improve penetration into cells by increasing their bioavailability. The functionalization of PtNPs by various molecules possessing a therapeutic effect opens up new possibilities for nanomedicine [214]. Thus, biosynthesized platinum nanoparticles represent a promising, economical drug for the disease treatment of various etiologies and improving the quality of people’s lives, requiring further study on their physicochemical and biological properties. Further knowledge of the synthesis mechanisms and the impact of PtNPs on the cells of living organisms will help to expand our understanding about them and discover new areas for their application.

## Figures and Tables

**Figure 1 jfb-13-00260-f001:**
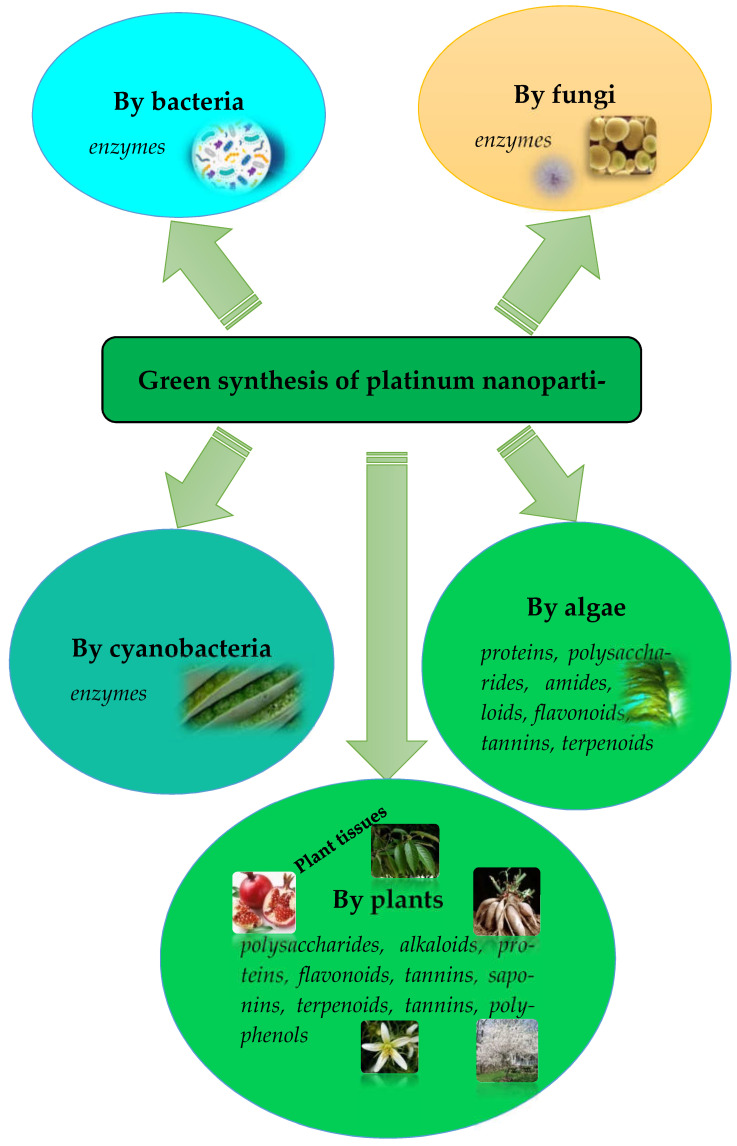
Biological synthesis of platinum nanoparticles (The molecules involved in platinum reduction are highlighted in italics).

**Figure 2 jfb-13-00260-f002:**
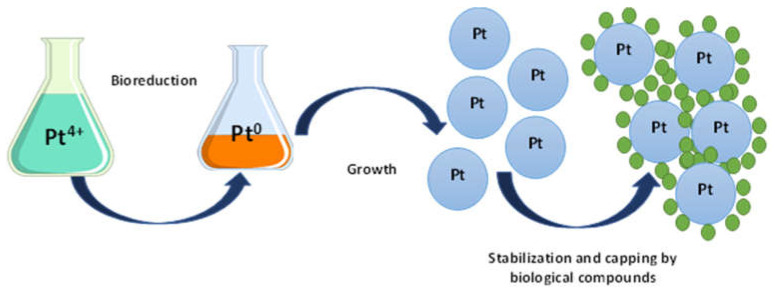
The proposed mechanism of platinum nanoparticles synthesis.

**Figure 3 jfb-13-00260-f003:**
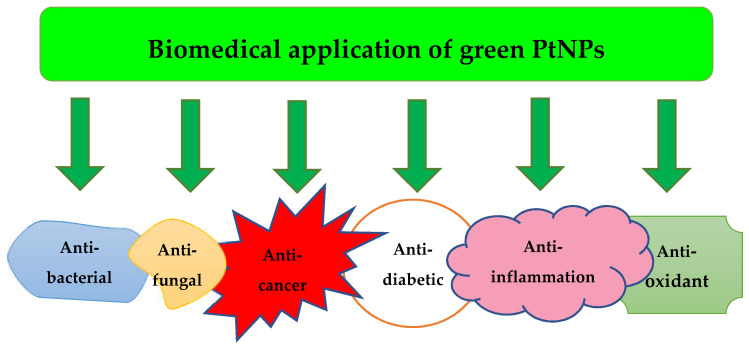
The main areas of PtNPs application.

**Figure 4 jfb-13-00260-f004:**
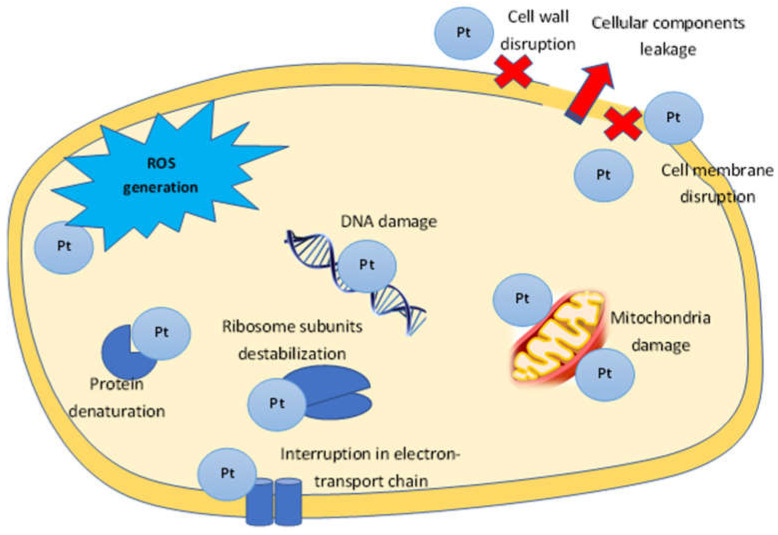
The proposed mechanism of PtNPs antibacterial activity.

**Figure 5 jfb-13-00260-f005:**
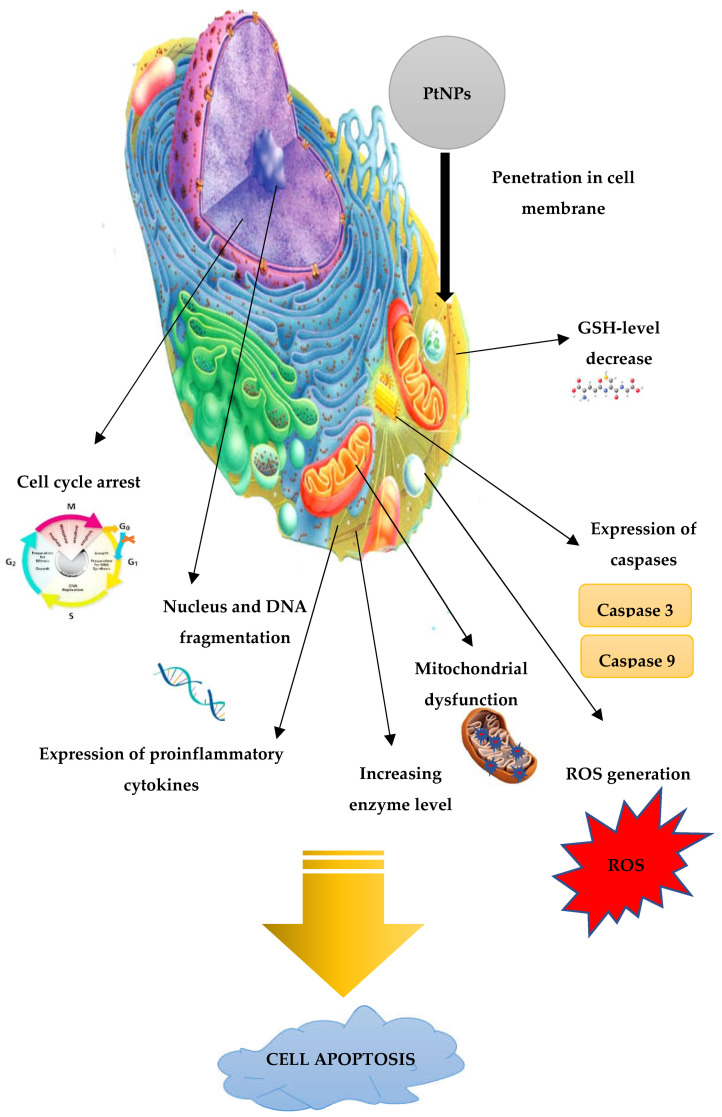
The proposed mechanism of PtNPs anti-cancer activity.

**Table 1 jfb-13-00260-t001:** Capping agents of the plant-mediated platinum nanoparticles.

Plant	Part of the Plant	Capping Agents	Reference
*Cacumen platycladi*	leaf extract	reducing sugars and flavonoids	[71]
*Punica granatum*	peel extract	polyphenols (ellagic acid, gallic acid, and quercetin)	[73]
*Barleria prionitis*	leaf extract	alkaloid, flavonoids, saponins, tannin, steroid, terpenoids, sterol, phenolic compound, glycosides	[74]
*Cochlospermum gossypium* (gum kondagogu)	aqueous medium containing gum	amino acids	[78]
*Quercus glauca*	leaf extract	flavonoids, tannins, carboxyl, amino and glycosides	[79]
*Ocimum sanctum*	leaf extract	ascorbic acid, gallic acid, terpenoids, proteins and amino acids	[80]
*Antigonon leptopus*	leaf stem and root extract	polysaccharides and proteins	[81]
*Azadirachta indica*	leaf extract	terpenoid	[75]
*Terminalia chebula*	fruit extract	polyphenolic content	[79]
*Bacopa monnieri*	leaf extract	amines, alcohols, ketones, aldehydes, and carboxylic acid	[85]
*Dioscorea bulbifera*	tuber extract	saponins, reducing sugars, ascorbic acid, citric acid, phenolics, and flavonoids	[86]
*Peganum harmala*	seed alkaloid fraction	alkaloid	[87]
*Salix tetraspeama*	leaf extract	polyphenols, polysaccharides, tannins, proteins, terpenoids	[88]
*Atriplex halimus*	leaf extract	glycosides, flavonoids, phenolic acids, and alkaloids	[89]
Date plants (Barni and Ajwa)	fruit extract	flavanols (polyphenols)	[90]
*Xanthium strumarium*	leaf extract	hydroxyl group	[91]
*Fumariae herba*	leaf extract	alkaloids (protopine), flavonoid compounds (quercetin-3,7-diglucoside, 3-arabinoglucoside, quercetin or rutin), and phenolic acids (p-coumaric acid, sinapic acid)	[92]
*Anacardium* *occidentale*	leaf extract	proteins, tannins, terpenoids, alkaloids, flavanols, phenols and glycosides	[93]
*Mentha piperita*	leaf extract	polyphenols	[94]
*Tragia involucrata*	leaf extract	polyphenols, alkaloids, flavonoids, and proteins	[95]
*Polygonum salicifolium*	leaf extract	glycosides, terpenoids, flavonoids, and alkaloids	[96]
*Prunus × yedoensis*	Gum extract	alkenes, alcohols, flavonoids, and amines	[98]
*Eichhornia crassipes*	leaf extract	polysaccharides	[99]
*Sapindus mukorossi*	aqueous extract of Soap nuts	saponins and flavonoids	[100]
*Centella asiatica*	leaf extract	flavonoids	[35]
*Combretum erythrophyllum*	leaf extract	proteins, flavonoids, amino acids, polyphenols, and carbohydrates	[101]
*Ononidis radix*	extract	isoflavonoids, phenolic acids	[102]
*Gloriosa superba*	tuber extract	alcoholic and phenolic compounds	[103]

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
