# Peer review of "Green Synthesis of Platinum Nanoparticles for Biomedical Applications"

_jfb, 2022, doi:10.3390/jfb13040260_

Round 1

Reviewer 1 Report

This review explored the bio-synthesis peculiarities of platinum nanoparticles (Pt NPs) based on living organisms using. The participation of various biological compounds in the Pt NPs synthesis is highlighted. Moreover, the biological activities, the proposed mechanisms and the potential biomedical applications of Pt NPs are discussed. This review isvaluable for publication in Journal of Functional Biomaterials after the authors addressing the following concerns.

1. The title of the manuscript should be changed to "Green synthesis of platinum nanoparticles for biomedical applications" to better cover the content of the article and attract readers of the journal.

2. The following paper related to the green synthesis of biomaterials need to be cited in the introduction. (Qi, C. et al. Biomolecule-assisted green synthesis of nanostructured calcium phosphates and their biomedical applications. Chem. Soc. Rev., 2019, 48, 2698-2737)

3. There is an article on a similar topic (Fahmy, S. A. et al. Platinum Nanoparticles: Green Synthesis and Biomedical Applications. Molecules 2020, 25, 4981), so the author needs to compare the differences between this article and this manuscript.

4. There are only two figures in this manuscript. The authors need to provide more figures to help readers understand, including their own figures.

5. It is suggested to summarize the green synthesis and the biomedical applications of Pt NPs into two tables, respectively.

Author Response

Dear colleague,

Thank you very much for your attentive attitude to my work and for the comments you made! I agree with all the comments, which are very valuable!

According to your remarks the following corrections were made (highlighted in green):

  1. The title of the manuscript was changed to "Green synthesis of platinum nanoparticles for biomedical applications".
  2. The article you kindly advised us was cited in the introduction (Qi, C. et al. Biomolecule-assisted green synthesis of nanostructured calcium phosphates and their biomedical applications. Chem. Soc. Rev., 2019, 48, 2698-2737).
  3. The article on a similar topic (Fahmy, S. A. et al. Platinum Nanoparticles: Green Synthesis and Biomedical Applications. Molecules 2020, 25, 4981) is an excellent paper devoted to green synthesis of platinum nanoparticles. However, it seems to me that the bias in the article is made mainly on the synthesis conditions (temperature, pH, etc.), the shape and size of nanoparticles. At the same time, there are enough examples of various applications described there. In the current article, we have tried to summarize the available information on the synthesis mechanism of PtNPs in biological objects, as well as the mechanisms of antibacterial, anticancer and other types of effects. All methods of biosynthesis are divided into separate subsections, as well as various applications of platinum nanoparticles. In addition, over the past 2 years, new publications on this topic have appeared, allowing for a broader coverage of the synthesis mechanism and the potential platinum nanoparticles application.
  4. Three new pictures (figures 1, 3 and 5) were added to the text to provide a better understanding of the article's issues and clarity.
  5. One of the added figures (figure 1) summarizes information about the objects of biosynthesis and the bio-compounds responsible for the synthesis of PtNPs. Another added figure (figure 1) provides information on the main directions of practical PtNPs application. It seems to me such a representation is the most visual and summarizing. I hope you find this acceptable, but if it is not, I will try to present the data in a tabular version.

Reviewer 2 Report

A very nice, extensive and complete review on a very important topic. It covers the major important subjects of the Pt NP green synthesis. It is an updated and important subject. It can be accepted as it is.

Author Response

Dear colleague,

Thank you very much for your attentive attitude to my work and for the kind response! According to the comments of other reviewers, some changes were made to the article, as well as 3 new figures were added. English has been corrected, .

Reviewer 3 Report

In this review, the author summarized the green synthesis of platinum nanoparticles. This review with a detailed summary of the present research status, including synthesis mechanism, and applications. I would recommend the acceptance of this review after minor revisions. Here, there are some issues that should be addressed and we give some suggestions for this review article as followed:

1-      The examples of biosynthesis are not well organized. It is unclear how the authors selected the examples in the review, and how these examples are ordered in such a way.

2-      More figures need to be added. It would be helpful if the authors can use some figures from the references so that the readers can get a better idea of the examples.

3-      English editing is recommended.

4-      The authors should add the important experimental published papers for the Biogenic synthesis of nanoparticles including the below-mentioned references in their revised manuscript:

-          ACS Omega 2020, 5, 43, 27811–27822

Author Response

Dear colleague,

Thank you very much for your attentive attitude to my work and for the comments you made! I agree with all the comments, which are very valuable!

According to your remarks the following corrections were made (highlighted in green):

  1. All examples of platinum nanoparticle biosynthesis were selected based on the problems of the article. We have tried to cover in the review the maximum number of articles devoted to the "green" synthesis of platinum nanoparticles, especially those that have been published over the past 10 years. The paper presents examples of biosynthesis using various types of living organisms, they were selected based on the production object (plants, microorganisms, etc.) and systematized in such a way for highlighting the synthesis mechanism as widely as possible. Particular importance was attached to the biocompounds that are involved in the platinum ion reduction, and capping agents that can potentially enhance the effect of PtNPs. Application examples of platinum nanoparticles were based on the activities shown by PtNPs and were considered in the light of their potential use.
  2. Three pictures (figures 1, 3 and 5) were added to the text to provide a better understanding of the article's issues and clarity. Unfortunately, it is quite difficult to use figures from the other papers due to copyright. We have tried to summarize all the necessary information in our own figures.
  3. We have tried to improve English based on the edits of our colleague, who knows English very well.
  4. The article you kindly advised us was cited in the introduction (ACS Omega 2020, 5, 43, 27811–27822).

Round 2

Reviewer 1 Report

The authors of the above-referenced manuscript have adequately answered the referees' questions and carefully revised the manuscript. The current revision is a better, improved and readable version, and it can be published without any changes.